# Low-Latency Semantic Processing with Optimal Prompt-Level Batching

Jingyi Qu
MIT
joycequ@mit.edu

Samuel Madden
MIT
madden@csail.mit.edu

Tianyu Li
University of Wisconsin-Madison
litianyu@cs.wisc.edu

## ABSTRACT

Semantic data processing systems use Large Language Models (LLMs) to unlock powerful semantic analysis over unstructured text, but often with high latency cost. In this paper, we measure and analyze execution latency of simple semantic operators over five different models and five workloads. Our analysis shows that LLM calls over public APIs have a surprisingly high per-call fixed overhead, which easily dominates token processing cost for many workloads. Using this insight, we analytically derive an optimal size for in-prompt document batching to effectively amortize this overhead, cutting end-to-end semantic operator latency by up to 14× with no meaningful accuracy loss.

**VLDB Workshop Reference Format:**
Jingyi Qu, Samuel Madden, and Tianyu Li. Low-Latency Semantic Processing with Optimal Prompt-Level Batching. VLDB 2026 Workshop: NOVAS.

**VLDB Workshop Artifact Availability:**
The source code, data, and/or other artifacts have been made available at https://github.com/joycequu/prompt-batching.git.

## 1 INTRODUCTION

Modern data systems increasingly rely on Large Language Model (LLM)-powered semantic operators [8, 11, 13] to process unstructured text. While powerful, semantic operators are slow: systems like Palimpzest [8] and Lotus [11] report hundreds of seconds of wall-clock time to process datasets of even moderate size. This latency often limits the usefulness of semantic operators in practical data processing pipelines [1, 5, 12].

A natural response to high latency is *prompt-level batching* – packing multiple documents into a single LLM call. Existing systems largely avoid this optimization for simple, tuple-independent operators (i.e., filter and map). The reason is two-fold. First, prompt-level batching is assumed to cost accuracy due to cross-document interference within a single prompt, and long-context degradation (i.e., "lost in the middle" [9]). Second, prompt-level batching may degrade, instead of improve, performance; due to the quadratic nature of attention computation [14], documents (and output) that share a prompt must attend to each other, which is expensive wasted computation. Consequently, most semantic processing systems default

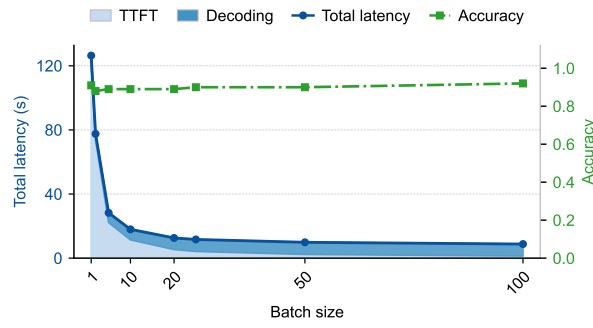

**Figure 1: Total latency and accuracy vs. batch size for a semantic filter for positive sentiment over 100 random movie reviews from SemBench on Gemini-3-Flash.**

to single document prompts and treat prompt-level batching as a high-risk, low-reward optimization best avoided.

In this paper, we show that neither reason holds in practice for filter and map operators. As shown in Figure 1, on a SemBench [7] filter over 100 movie reviews, end-to-end completion latency collapses to around 7% of the one-document-per-prompt baseline, while accuracy holds flat at 0.9. For accuracy, we believe this is because modern LLMs are powerful enough to judge each packed document independently and minimize interference when clearly prompted. For latency, we analyze our performance measurements to reveal a high per-call fixed cost (e.g., queuing, streaming model weights from memory) that dominates execution time on public APIs. Prompt-level batching is a clear win because it amortizes this cost. Using a decomposed latency model of LLM serving engines, we further show that most semantic operator workloads can be batched enough to amortize this fixed overhead while still operating within a "linear band" where attention's quadratic cost remains negligible. From this model we derive a closed-form solution for the optimal prompt-level batching scheme given a query workload and model, reducing end-to-end latency across semantic operator benchmarks by 1.5–14× with no measurable accuracy loss. In contrast, previous systems achieve comparable speedups by controlling fine-grained serving-engine scheduling (Lotus [11]) or exploiting massive parallelism (Palimpzest [8]); prompt-level batching is a simple yet effective way to replicate these gains when running semantic operators on rate-limited public-facing APIs—arguably the more common setting. In summary, the main contributions of this paper are:

(1) We show that prompt-level batching yields large latency reductions with no accuracy loss.

| Dataset | Type | Semantic Operator | Input | Output | Difficulty |
|---|---|---|---|---|---|
| SemBench (Movie Reviews) | sem_filter | Filter movie reviews: *"Is this review clearly positive?"* (~135 tokens) | short (~50 tokens) | short (~17 tokens) | easy (binary sentiment) |
| TweetEval (Irony) | sem_filter | Filter tweets: *"Does this tweet use irony or sarcasm?"* (~135 tokens) | short (~40 tokens) | short (~15 tokens) | medium (implicit irony) |
| LitSearch (Scientific Papers) | sem_filter | Filter NLP/AI paper abstracts: *"Are there any research papers on methods to compress large-scale language models using task-agnostic knowledge distillation techniques?"* (~130 tokens) | medium (~210 tokens) | short (~21 tokens) | medium (sparse retrieval) |
| TweetEval (Emotion) | sem_map | Map tweets to emotion: *"Which emotion is expressed?"* (anger, joy, optimism, sadness) (~200 tokens) | short (~40 tokens) | short (~15 tokens) | medium (4-class) |
| CUAD (Legal Contracts) | sem_map | Map commercial contracts: extract values for 41 pre-defined legal clause types (parties, dates, obligations, etc.) (~2,070 tokens) | long (~9800 tokens) | medium (~960 tokens) | hard (41 legal fields) |

**Table 1: Datasets and semantic operators in our task sweep, with approximate per-document input/output token counts.**

(2) We present a decomposed linear latency model that fits observed behavior with high fidelity and isolates per-call overhead as the dominant cost.

(3) We derive a closed-form analytical solution for the recommended batch size given workload signature and model information

## 2 EXPERIMENTS

### 2.1 Experimental Setup

To demonstrate that the latency and accuracy phenomena we observe in Figure 1 generalize across general tasks and closed-source, provider-based APIs, we ran a benchmark over commercial APIs through OpenRouter. We selected five distinct tasks (Table 1) spanning multiple semantic operator types (filters and maps), sequence lengths, and reasoning difficulties. These tasks range from simple binary sentiment classification on short sequences (SemBench [7]) to dense, structured extraction over massive context windows (CUAD [4]).

We evaluated these workloads across a diverse set of models spanning different providers and capability tiers: Gemini-3-Flash, GPT-5.4, GPT-5.4 Nano, Claude 4.5 Haiku, and Claude 4.6 Sonnet. For all benchmark runs, internal reasoning features (extended chain-of-thought traces) were explicitly disabled. This keeps output length and performance predictable. Furthermore, to ensure that the measured latency reductions were strictly the result of multi-document batch processing rather than provider-side optimizations, we explicitly disabled prefix caching by prepending a unique random nonce to every API request.

For each dataset and model combination, we swept batch sizes ranging from $b = 1$ up to $b = 100$ (or the maximum context limit, whichever was reached first). For a target workload of 100 documents, we issued $\lceil 100/b \rceil$ independent LLM calls for each batch size. All baseline evaluations were standardized: we use bulleted lists for tuple separation with the operator instructions placed before the packed tuples.

### 2.2 Latency collapses with batch size

Our experiments reveal substantial latency reductions from prompt-level batching. Aggregating multiple documents into a single prompt drastically reduces wall-clock time, and the effect holds across providers and different sizes of models.

Figure 2 reports total latency against batch size for each (model, dataset) pair. For the four short-output tasks (SemBench, TweetEval Irony, TweetEval Emotion, LitSearch), every model exhibits a steep drop between $b = 1$ and $b \approx 10$, followed by a gradual flattening for large $b$. The collapse is concentrated in TTFT, which at $b = 1$ accounts for nearly all of the total latency and shrinks rapidly as fewer calls are issued. Decode remains roughly flat across batch sizes and contributes a small share throughout. On Gemini-3-Flash, SemBench, LitSearch and TweetEval Irony exceed 100 seconds at $b = 1$ and fall to around 10 seconds at $b = 100$, a 14× reduction. Other models exhibit comparable order-of-magnitude collapses on these tasks. This consistency across providers indicates the underlying cost structure is shared across providers and models.

The reduction is on a smaller magnitude for longer-output tasks. CUAD's 960-token outputs make decode the dominant component at every batch size, so the TTFT collapse contributes a smaller share of total latency and the curve declines more gradually. The per-document input and output determines how much batching reduces total latency, and we formalize this in Section 3 through latency decomposition.

### 2.3 Correctness robust to batching

The primary hesitation in adopting multi-document batching in existing systems stems from concerns over accuracy degradation. Our evaluation indicates that for modern LLMs executing standard declarative data processing tasks, this concern does not hold in practice.

As shown in Figure 2, for both semantic filters (SemBench, TweetEval Irony) and semantic maps (TweetEval Emotion), accuracy remains strictly flat as batch sizes scale from $b = 1$ up to $b = 100$. Under GPT-5.4 and Claude Haiku 4.5, LitSearch exhibits visible fluctuations across batch sizes, but this reflects metric instability

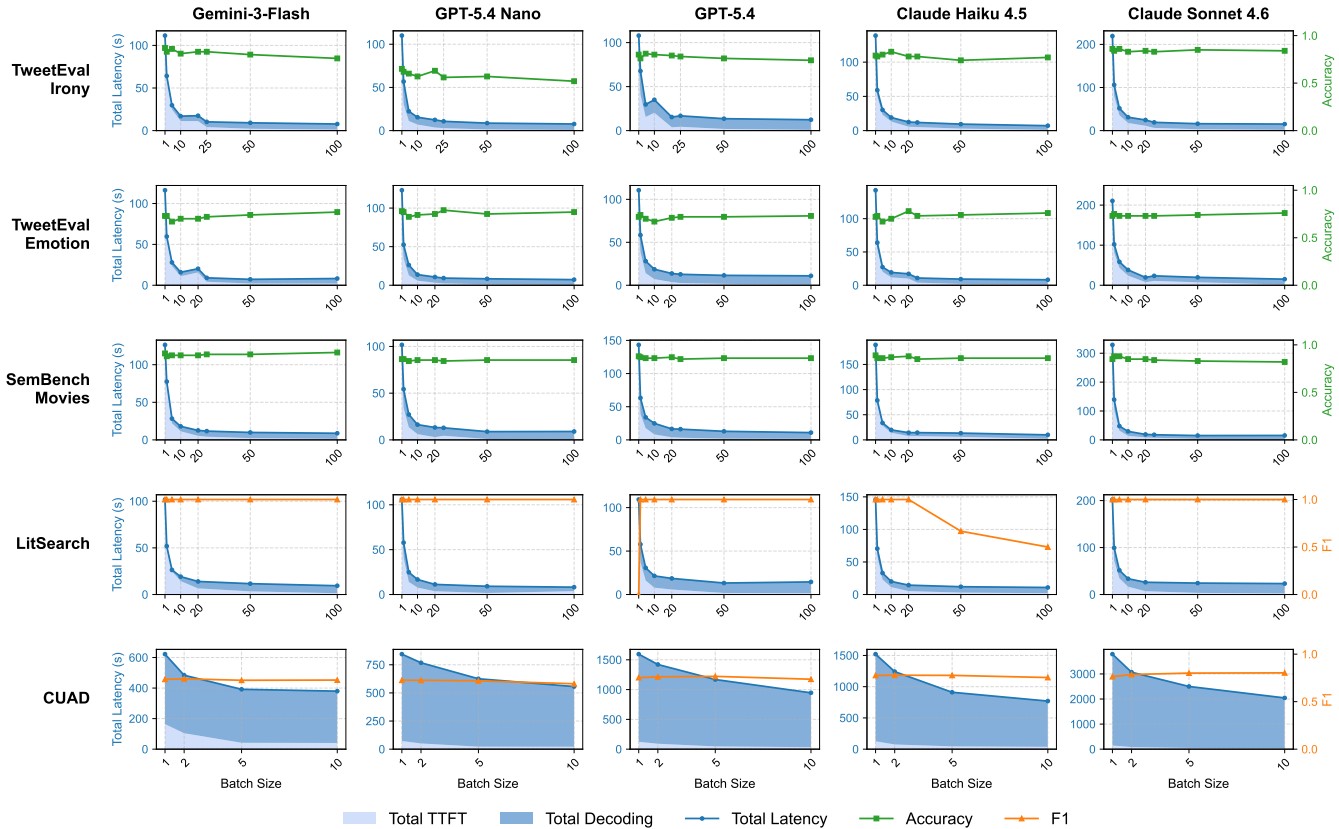

**Figure 2: Total latency (left axis) and accuracy or F1 (right axis) vs batch size for each (model, dataset) pair.**

rather than batching effects as only one of the 100 documents is relevant. Even for the dense, structured extraction required by the CUAD dataset, which pushes the prompt toward 10,000 tokens, the F1 score remains stable across batch sizes. Overall, the model does not become confused by the presence of other tuples in its context window.

To ensure robustness is not an artifact of prompt engineering, we tested the models' sensitivity to prompt structure. Neither the delimiter (JSON, Markdown, or bullet points) nor instruction order (operator before vs. after the documents) had a statistically meaningful effect on latency or accuracy across any model tested. Hence, we can conclude that for typical declarative data pipeline workloads, there is no meaningful tradeoff between latency and accuracy. Multi-document batching provides massive latency reductions with no measurable correctness loss.

## 3 ANALYSIS

### 3.1 The Decomposed Linear Latency Model

We observe that most runs in Figure 2 trace a hyperbola in the batch size $b$: a steep drop into a flat tail. Since the call count is $\lceil D/b \rceil$, where $D$ is the number of documents, a drop that tracks $1/b$ is the signature of a fixed per-call cost – independent of input and output size – amortized across the batch. As $b$ grows this cost is amortized

away and latency settles onto a floor of pure per-document processing. Moreover, this floor largely stays constant and does not rise as batch sizes increase. Recall that when batching, the total input and output tokens are essentially invariant to $b$ as the same documents are processed regardless of packing; only the system prompt is shared within a batch. This suggests that batching triggers no quadratic blow-up, and the per-document cost is essentially the same whether the workload is spread across many small calls or packed into one large call. We therefore approximate the latency of simple semantic operators with a linear model, separating the two phases of LLM forward pass [16]. For a single call, let $N_{\text{in}}$ and $N_{\text{out}}$ denote the per-call input and output token counts:

$$\text{Per-call Prefill Time} \approx \beta_0 + \beta_1 \cdot N_{\text{in}} \tag{1}$$

$$\text{Per-call Decode Time} \approx \beta_2 \cdot N_{\text{out}} \tag{2}$$

$$\text{Latency} \approx \beta_0 + \beta_1 \cdot N_{\text{in}} + \beta_2 \cdot N_{\text{out}} \tag{3}$$

Here, $\beta_0$ is the fixed per-call overhead (queuing, scheduling, network round-trip), and $\beta_1, \beta_2$ are the marginal latencies per input (prefill) and output (decode) token respectively. Aggregating over the workload – whose total document tokens $n_{\text{in}}, n_{\text{out}}$ (excluding the system prompt $N_{\text{sys}}$, which is repeated once per call) are essentially invariant to $b$ – predicted latency is

$$\text{Overall Latency} \approx \left\lceil \frac{D}{b} \right\rceil (\hat{\beta}_0 + \hat{\beta}_1 N_{\text{sys}}) + \hat{\beta}_1 n_{\text{in}} + \hat{\beta}_2 n_{\text{out}}, \tag{4}$$

| Model | Dataset | $\hat{\beta}_0$ (sec) | $\hat{\beta}_1$ (sec/token) | $\hat{\beta}_2$ (sec/token) | $R^2$ |
|---|---|---|---|---|---|
| **Gemini-3-Flash** | SemBench Movies | 1.041330 | 0.000043 | 0.004346 | 0.967 |
| | TweetEval Irony | 1.152903 | 0.000000 | 0.004425 | 0.983 |
| | TweetEval Emotion | 0.884882 | 0.000198 | 0.003901 | 0.969 |
| | LitSearch | 1.137858 | 0.000020 | 0.003765 | 0.906 |
| | CUAD | 1.576799 | 0.000008 | 0.003805 | 0.840 |
| **GPT-5.4 Nano** | SemBench Movies | 0.679805 | 0.000000 | 0.006930 | 0.957 |
| | TweetEval Irony | 0.620145 | 0.000034 | 0.006018 | 0.867 |
| | TweetEval Emotion | 0.576811 | 0.000000 | 0.006550 | 0.770 |
| | LitSearch | 0.389395 | 0.000140 | 0.004055 | 0.596 |
| | CUAD | 0.290724 | 0.000025 | 0.006942 | 0.963 |
| **GPT-5.4** | SemBench Movies | 0.803740 | 0.000142 | 0.008090 | 0.893 |
| | TweetEval Irony | 0.733995 | 0.000028 | 0.009817 | 0.945 |
| | TweetEval Emotion | 0.712661 | 0.000000 | 0.009801 | 0.971 |
| | LitSearch | 0.655615 | 0.000037 | 0.009655 | 0.961 |
| | CUAD | 1.120737 | 0.000016 | 0.009638 | 0.974 |
| **Claude Haiku 4.5** | SemBench Movies | 1.283587 | 0.000304 | 0.004766 | 0.957 |
| | TweetEval Irony | 1.084030 | 0.000000 | 0.003748 | 0.967 |
| | TweetEval Emotion | 1.059233 | 0.000000 | 0.004512 | 0.952 |
| | LitSearch | 0.990082 | 0.000041 | 0.005531 | 0.931 |
| | CUAD | 1.167197 | 0.000023 | 0.006860 | 0.780 |
| **Claude Sonnet 4.6** | SemBench Movies | 1.673606 | 0.000053 | 0.009120 | 0.760 |
| | TweetEval Irony | 1.516723 | 0.000026 | 0.009104 | 0.935 |
| | TweetEval Emotion | 1.311519 | 0.000140 | 0.010953 | 0.909 |
| | LitSearch | 1.367250 | 0.000035 | 0.014789 | 0.968 |
| | CUAD | 0.760575 | 0.000028 | 0.016888 | 0.927 |

Table 2: Fitted latency model parameters $\hat{\beta}_0$, $\hat{\beta}_1$, $\hat{\beta}_2$, and goodness-of-fit $R^2$ for each (model, dataset) pair, obtained by NNLS on Eq. 3.

the hyperbola we observed. We validate this against observed latency using $R^2$.

We fit these parameters by sweeping over batch sizes $b$. For a workload of $D$ documents, we issue $\lceil \frac{D}{b} \rceil$ independent LLM calls. At each $b$, we extract per-call TTFT, decode time, and token counts, and apply non-negative least squares (NNLS) to both regression stages, enforcing the physically motivated constraint that time and computation cannot be negative. This yields the estimated parameters: $\hat{\beta}_0$, $\hat{\beta}_1$, and $\hat{\beta}_2$, which we validate against observed overall latency using $R^2$.

Table 2 shows our result. With $R^2$ values largely above 0.9, the model predicts latency accurately. The fitted parameters reveal three findings.

*(1) Marginal prefill latency is negligible ($\beta_1 \approx 0$).* Across all five models and five tasks, the fitted $\beta_1$ values are at most $3 \times 10^{-4}$ sec/token and frequently zero under the non-negative constraint.

*(2) Decode time per token ($\beta_2$) is a stable constant per-model.* The decode time per output token $\beta_2$ remains stable within a given model family, regardless of the underlying semantic task. Gemini-3-Flash maintains a $\beta_2 \approx 0.004$ sec/token whether classifying sentiment or extracting legal clauses; GPT-5.4 sits at 0.008-0.011; Claude Sonnet 4.6 at 0.009-0.016. This per-model stability is what makes one-time calibration sufficient.

*(3) Per-call overhead dominates ($\beta_0$).* The fixed per-call overhead ranges from 0.39 to 1.67 seconds across providers, reflecting server-side scheduling, network round-trips, and fixed cost (e.g., streaming model weights from memory). For short-tuple tasks, this overhead constitutes the majority of execution time. At $b = 1$, a 100-document SemBench workload on Gemini-3-Flash spends roughly 100 seconds simply incurring per-call overhead (Figure 3) without doing per-token work.

## 3.2 Analytical Bounds on the Linear Regime

We now ground the linear model of Section 3.1 in hardware, deriving where linearity holds and where it breaks. A modern GPU's compute units act on data staged on fast on-chip SRAM, fed by slow High-Bandwidth Memory (HBM). To understand the latency behavior of modern GPUs under different workloads, we must first understand the *arithmetic intensity* of each workload. Arithmetic intensity is defined as FLOPs performed per byte moved between HBM and SRAM. With peak compute throughput $\pi$ (FLOP/s) and HBM bandwidth $\rho$ (B/s), wall-clock time $T$ is bounded by $T \geq \max(\text{FLOPs}/\pi, \text{Bytes}/\rho)$. These two terms cross at a "ridge point" $I^* = \pi/\rho$ [15]. Below $I^*$ a kernel is memory-bound and latency is determined by bytes moved; above $I^*$ it is compute-bound and latency is determined by FLOPs performed. As mentioned, LLM

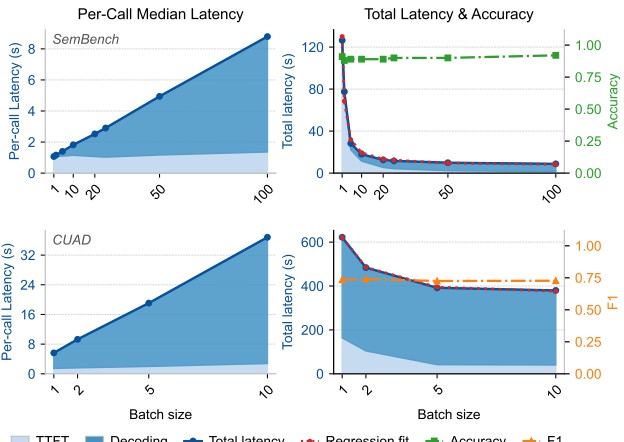

**Figure 3: Per-call median latency (left) and aggregate workload latency with regression fit (right) for Gemini-3-Flash on SemBench (top) and CUAD (bottom). Detailed regression fit in Table 2.**

serving comprises two phases with very different intensities—prefill and decode. In the remainder of this section, we locate each relative to $I^*$ separately. We find that throughout the regime where semantic operators run, prefill is approximately linear in $N_{\text{in}}$ and decode is approximately linear in $N_{\text{out}}$, recovering the calibrated form of Eq. 4 from first principles.

*3.2.1 Prefill Analysis.* Prefill is the forward pass that ingests the prompt and populates the KV cache before the first output token is decoded. For simplicity, we assume that the entire prompt is processed in a single pass for now. Intuitively, then, the work, in FLOPS, done by each transformer block decomposes into a linear term covering the projection matrices and MLP (each token processed against fixed weights) and a quadratic term covering self-attention (pairwise interactions across the sequence) [6, 10]:

$$C(N) = \underbrace{2\,P_b\,N}_{C_{\text{lin}}^{\text{block}}(N)} + \underbrace{2\,d_{\text{model}}\,N^2}_{C_{\text{attn}}^{\text{block}}(N)} \quad (5)$$

where $d_{\text{model}}$ is the model width and $P_b$ the number of transformer-block parameters. Every transformer block in a depth-$L$ stack has identical structure and processes the same $N$ tokens, so the full-pass cost is $L$ times the per-block cost.

With modern IO-aware kernels such as FlashAttention [2], the attention matrix operation is tiled into SRAM and does not materialize. The overall memory overhead of the block is:

$$M(N) = \underbrace{s\,P_b}_{\text{weights}} + \underbrace{2s\,d_{\text{model}}\,N}_{\text{kv cache written}} \quad (6)$$

where $s$ is bytes per parameter in the model. To understand the latency behavior, we must understand when the bottleneck shifts from memory to compute, i.e. where $\frac{C(N)}{M(N)} = I^* = \frac{\pi}{\rho}$. Then, we can compute:

$$\begin{aligned}
\frac{C(N)}{M(N)} &= \frac{2P_b N + 2\,d_{\text{model}}\,N^2}{sP_b + 2s\,d_{\text{model}}\,N} \\
&= \frac{2P_b N}{sP_b} \cdot \frac{1 + d_{\text{model}}N/P_b}{1 + 2\,d_{\text{model}}N/P_b} \\
&= \frac{2N}{s} \cdot \frac{1 + d_{\text{model}}N/P_b}{1 + 2\,d_{\text{model}}N/P_b}. \quad (7)
\end{aligned}$$

The leading factor $2N/s$ is independent of $P_b$, so the crossover does not depend on model size. Observe that $P_b \approx 12d_{\text{model}}^2$, dominated by weight matrices in the attention and MLP layer. For small $N$, this makes $sP_b \gg 2d_{\text{model}}N$ as $d_{\text{model}}$ is typically large (e.g., 4096), $d_{\text{model}}N/P_b \approx 0$, and $C(N)/M(N) \approx 2N/s$. Setting this equal to the ridge point $I^* = \pi/\rho$ gives the memory-to-compute crossover:

$$\frac{2N}{s} = \frac{\pi}{\rho} \quad \Longrightarrow \quad N_1^* = \frac{I^* s}{2} = \frac{\pi s}{2\rho}. \quad (8)$$

For an H200 GPU and FP8 parameters, this comes out to $N \approx 200$. However, this number assumes that the compute node performs a full forward pass on the MLP per block, which is not true for modern Mixture-of-Experts architectures [3]. Under MoE, a router sends each token to the top $k$ "experts" (smaller MLP) to process. This introduces a sparsity factor into each compute, as only a fraction of $P_b$ is involved, but over a long sequence, the model still may need to stream a large portion (if not all) model weights. For a model like DeepSeek-V3, this can lead to a factor of approximately 20× reduction in compute, leading to $N \approx 4000$ before compute becomes the bottleneck, which explains the observed linear scale up for workloads like SemBench and TweetEval.

However, workloads like LitSearch and CUAD, as predicted by this model, would be in the quadratic region despite a good linear fit. This is because compute-induced latency is $C(N)/\pi$, which means the constant factor for the quadratic term is $2Ld_{\text{model}}/\pi$, a vanishingly small value with $\pi \approx 10^{15}$ for an H200, and $2Ld_{\text{model}}$ in the range of $10^4$ to $10^5$. The linear model fits $\beta_1 \approx \frac{2P_b + 2d_{\text{model}}N}{\pi}$, which, until $N$ is large enough to outstrip $\pi$, is but a rounding error, which explains the empirical linear fit and the near-0 $\beta_1$ fit.

*3.2.2 Decode Analysis.* Decode is the stage where the model is autoregressively generating output tokens. Unlike prefill, decoding is strictly sequential and is always memory-bound. Its latency for each transformer block on an input sequence of length $N$ is:

$$M_{decode}(N) = \underbrace{s\,P_b}_{\text{weights}} + \underbrace{2s\,d_{\text{model}}\,N}_{\text{kv cache read}} + \underbrace{2s\,d_{\text{model}}}_{\text{kv cache written}} \quad (9)$$

Summing per-step traffic across prompt length $N_{\text{in}}$ and $N_{\text{out}}$ steps:

$$\begin{aligned}
M_{\text{total}} &= \left(sP_b + 2sd_{\text{model}}N_{\text{in}}\right)N_{\text{out}} + \sum_{n=0}^{N_{\text{out}}-1} 2sd_{\text{model}}\,n \\
&= \left(sP_b + 2sd_{\text{model}}N_{\text{in}}\right)N_{\text{out}} + 2sd_{\text{model}} \cdot \frac{N_{\text{out}}(N_{\text{out}} - 1)}{2} \\
&= \left[sP_b + sd_{\text{model}}\left(2N_{\text{in}} + N_{\text{out}} - 1\right)\right]N_{\text{out}}, \quad (10)
\end{aligned}$$

|  | Gemini 3-Flash | GPT 5.4 N. | GPT 5.4 | Claude Haiku 4.5 | Claude Sonnet 4.6 |
|---|---|---|---|---|---|
| TweetEval Irony | 82 | 89 | 88 | 88 | 89 |
| TweetEval Emotion | 72 | 92 | 89 | 88 | 90 |
| SemBench Movies | 87 | 89 | 94 | 90 | 94 |
| LitSearch | 68 | 91 | 94 | 88 | 91 |
| CUAD | 75 | 100 | 100 | 96 | 83 |

**Table 3: Latency reduction captured at $b = 8$, as a percentage of the reduction between $b = 1$ and the latency-minimizing batch size in the sweep.**

Importantly, for semantic filter and semantic map, $N_{out}$ is typically small in comparison to the input document (e.g., outputting a true/false for each document that is 1000s of tokens). Therefore, $2N_{in} + N_{out} - 1 \approx 2N_{in}$ for semantic operators, which gives the linear decode term. We note that this linearization is a slight simplification: from Eq. 10, per-token decode traffic grows with $N_{in}$ and therefore with $b$, so $\beta_2$ is not strictly constant across workloads and batch sizes. In practice, this effect is small relative to the $\beta_0$ overhead and is only visible on Claude Sonnet 4.6 in our sweep.

## 4 RECOMMENDED BATCH SIZE

Current relational LLM systems often propose learned models or adaptive runtime execution engines to find optimal batch sizes. Our findings suggest this is overkill: the recommended batch size is available in closed form. Eq. 4 shows that total latency strictly decreases in $b$ — only the per-call-overhead term $\frac{D}{b}(\beta_0 + \beta_1 N_{sys})$ depends on $b$, and it shrinks monotonically with no meaningful accuracy loss. The system-prompt contribution $\beta_1 N_{sys}$ within this term is further reduced toward zero by prefix caching, which most providers apply automatically once the number of input tokens exceeds a minimum length (e.g. 1024 tokens for GPT models). Batching lifts the per-call prompt past this threshold, so workloads that could not benefit from caching at $b = 1$ become cache-eligible at larger $b$. The dominant remaining cost is therefore the per-call overhead $\beta_0$. In principle, a system should batch as large as the context window and rate limits allow. In practice, the curve flattens quickly, and at large context window our approximation would become inaccurate. We therefore follow a rule of thumb: pick a batch size large enough to capture most of the available reduction.

We define the cutoff via the marginal latency saved per added document. Differentiating the $b$-dependent term of Eq. 4 and normalizing by $D$, then setting the result equal to a tolerance $\tau$ (seconds saved per added document), yields

$$\frac{\beta_0 + \beta_1 N_{sys}}{b^2} = \tau \quad \implies \quad b^* = \sqrt{\frac{\beta_0 + \beta_1 N_{sys}}{\tau}} \approx \sqrt{\beta_0/\tau}, \quad (11)$$

where the approximation uses $\beta_1 \approx 0$ from Table 2 (the $\beta_1 N_{sys}$ term contributes under 0.15s even for a 500-token system prompt, an order of magnitude below $\beta_0$).

Plugging the fitted $\beta_0$ values from Table 2 into Eq. 11 with $\tau = 0.02$ s/doc gives $b^*$ between 5.8 and 9.2 across all 25 (model, dataset) pairs, clustering around 8. We therefore recommend $b = 8$ as a default. Empirically, $b = 8$ recovers the bulk of the available latency reduction. Table 3 reports the fraction of latency reduction captured

at $b = 8$, defined as $(L_1 - L_8)/(L_1 - L_{min})$ where $L_{min}$ is the lowest observed latency in the sweep. Across all 25 measured (model, dataset) pairs, the median capture is 89% and 23 of 25 pairs exceed 75%. Thus, the constant $b = 8$ recommendation transfers across models, workloads, and operator types in our sweep.

## 5 RELATED WORK

Systems such as PALIMPZEST [8], LOTUS [11], and DocETL [13] predominantly issue one LLM call per document per operator. Whether to deviate from this default by batching multiple documents into a single prompt is treated as a difficult, task-dependent decision: batching can reduce cost, but may degrade accuracy through cross-tuple interference, with uncertain latency effects. Existing systems engage with batching at different level. PALIMPZEST executes with an optimizer that adapts to cost, runtime, and quality but inherits the relational iterator model, processing one record at a time[8] and thus is not built for packing multiple tuples into a single LLM call. LOTUS batches at the execution level rather than the prompt level, parallelizing many single-document calls rather than packing documents into a single prompt; for local execution, batched inference under vLLM amortizes per-call overhead across concurrent requests, achieving similar latency reductions. LOTUS reserves multi-tuple packing when necessary for operators where cross-document reasoning is required (sem_topk, sem_agg, and sem_join). DocETL [13] and SEMA [12] go further, treating prompt-level batching as a tunable parameter. DocETL sweeps candidates empirically because the optimum is task-dependent, and SEMA tunes its adaptively at runtime against a user-specified accuracy tolerance. Notably, both LOTUS and DocETL underline that inter-record packing is bounded above by long-context degradation, and both cite "lost in the middle" as evidence that quality degrades as more content is packed into a single prompt [9], while SEMA reports empirically that prompt batching consistently reduces token cost but can increase latency under local inference (where vLLM's scheduler favors many short prompts over fewer long ones) and can degrade quality through cross-tuple interference within a single prompt.

## 6 CONCLUSION

We showed that prompt-level batching delivers large latency reductions for filter and map operators with no meaningful accuracy loss. A decomposed linear latency model isolates per-call overhead as the dominant cost across five models and five workloads, and a hardware-grounded analysis confirms that for workloads in the semantic-operator regime, prefill and decode both fall within the linear band where attention's quadratic cost stays negligible. A fixed $b = 8$ captures roughly 89% of the available latency reduction across the workloads and models we tested. Thus, semantic data processing systems running against public APIs would benefit from batching by default.

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
