# OpenReview forum: "Low-Latency Semantic Processing with Optimal Prompt-Level Batching"
_VLDB.org/2026/Workshop/NOVAS — NOVAS 2026_

### Official Review · Reviewer_Aa48 · 2026-07-10

**Confidence:** 3

**Improvement Opportunities:**

W1. The main latency gain needs stronger baselines, especially parallel single-document calls.
The paper does not clearly state whether the independent LLM calls under a given batch size are executed serially or in parallel. If the baseline executes 100 single-document requests serially, then much of the latency reduction from batching is expected. To establish the real benefit of prompt-level batching, the paper should include a comparison with parallel single-document API calls.

W2. The “no meaningful accuracy loss” claim is too strong.
The paper uses only 100 documents per workload and lacks confidence intervals, repeated runs, parse failure rates, missing-output rates, and order-mismatch rates. Therefore, the current results only show that no obvious accuracy degradation was observed in the tested workloads, but they do not sufficiently support a more general “no meaningful accuracy loss” claim.

W3. The linear latency model should be framed as an empirical approximation rather than a general serving model.
From the perspective of Transformer computation, the prefill stage contains a quadratic self-attention term. Packing multiple documents into one prompt changes the attention cost from “the sum of squared document lengths” to “the square of the total document length,” which introduces cross-document attention cost. Therefore, the linear latency model should be presented as an empirical approximation within the tested regime, rather than as a general LLM serving model.

W4. The artifact link is unavailable.
The paper states that the code, data, and artifacts are publicly available, but the GitHub link provided in the paper currently returns a 404 error. Since the paper’s conclusions heavily depend on latency measurement, prompt construction, output parsing, and regression fitting details, the unavailable artifact substantially weakens reproducibility.

**Minor Comments:**

D1. The paper should clarify the execution model and add a stronger latency baseline.
The paper states that, for a workload of 100 documents and batch size b, it issues approximately 100/b independent LLM calls. However, it does not clarify whether these calls are executed sequentially or in parallel. This detail is crucial because prompt-level batching is most beneficial under serial execution or strong rate limits. If a practical system can send multiple single-document requests in parallel, the latency benefit of batching over a parallel baseline may be much smaller. I suggest that the authors compare at least three settings: serial single-document calls, parallel single-document calls, and prompt-level batching.

D2. The correctness evaluation should include system-level validity metrics.
Reporting only accuracy and F1 is insufficient for evaluating the reliability of prompt-level batching in semantic data systems. When multiple tuples are packed into the same prompt, the model may omit some outputs, generate extra outputs, reorder results, produce unparsable JSON, or allow one tuple to influence the judgment of another. I suggest reporting parse failure rate, missing tuple rate, order mismatch rate, and retry rate. These metrics are especially important for semantic operators, since data systems usually require stable and reliable structured outputs.

D3. The latency model is useful but its validity range should be stated more carefully.
The decomposed linear latency model is intuitive and useful, and it fits many experimental results. However, it is not a complete complexity model of Transformer inference. In theory, prefill includes quadratic attention cost with respect to input length, and decode also depends on both input length and generated length. The paper should more clearly state that the model is an empirical approximation for the public-API semantic-operator workloads tested in this paper. Stress tests with longer documents, larger batch sizes, and local serving would help validate the boundary of the model’s applicability.

D4. The artifact availability issue should be fixed.
The artifact repository listed in the paper is currently inaccessible. The authors should make the repository public or correct the link. The artifact should include at least raw latency logs, prompt templates, batch construction code, token accounting, output parsing scripts, regression fitting code, and figure generation scripts. This is particularly important because the main contribution of the paper relies on empirical measurement.

**Short Summary:**

This paper studies prompt-level batching for LLM-powered semantic data processing operators, mainly focusing on tuple-independent sem_filter and sem_map workloads. The key observation is that, when using public LLM APIs, the end-to-end latency of semantic operators is often dominated by a large fixed per-call overhead rather than purely by token processing cost. Therefore, packing multiple documents into a single prompt can reduce the number of API calls and amortize the fixed overhead, leading to substantial latency reduction. The paper evaluates five workloads across five commercial models, proposes a decomposed linear latency model, and further provides an empirical recommendation for the default batch size. Overall, the topic is timely and highly relevant to execution optimization for LLM-based semantic data processing systems, and it fits the NOVAS workshop theme well. However, several claims are stronger than the current evidence supports, especially regarding accuracy robustness, the generality of the linear latency model, and the boundary of batching benefits in public-API settings. I therefore lean toward weak accept, but I suggest that the authors clarify the scope and soften some of the claims.

**Strong Points:**

S1. The paper addresses a timely and relevant problem in LLM-powered semantic data processing.
The paper focuses on the latency problem of LLM-powered semantic operators, which is an important bottleneck in current semantic data processing systems. In particular, the paper studies the public-API setting, where users usually cannot control the underlying serving engine, making the problem practically meaningful.

S2. The main empirical observation is clear and practically useful.
The paper clearly shows that prompt-level batching can significantly reduce total latency for short-input and short-output semantic filter/map workloads. The main reason is that batching reduces the number of API calls and thus amortizes the fixed per-call overhead. Although this observation is intuitive, it still has direct value for practical system design.

S3. The evaluation covers multiple workloads and models.
The paper evaluates five semantic workloads across five commercial models, including sentiment filtering, irony detection, emotion classification, paper retrieval, and legal contract extraction. Compared with a single-model or single-task case study, this broader coverage makes the empirical observation more convincing.

S4. The proposed latency model is simple and actionable.
The decomposed latency model is approximate, but it intuitively explains why the latency curve changes with batch size. It also provides a simple batching strategy reference for system builders.

---

### Official Review · Reviewer_Fowr · 2026-07-10

**Confidence:** 4

**Improvement Opportunities:**

OoI:
- The authors say that contemporary work exploit "massive parallelism" for low query latency. How does
that interact with prompt-level batching? Is it an orthogonal optimization yielding to further gains. Please clarify.
- The authors say they disabled prefix-caching in their experiments. However, prefix caching can also lead to low-latency semantic queries, but prompt-level batching could make it harder to utilize. It would be good if the authors could elaborate the effect of larger batch sizes on prefix caching, especially when multiple semantic operators are executed in sequence. In this scenario, the LLM serving sytem needs to only compute the KV-cache for texts once, then reuse it for subsequent operators. Prompt-level caching might reduce the number of cache hits.
- The tasks the authors evaluted on are mainly rather simple tasks with short texts. What happens in cases with longer texts (e.g. multi-page pdfs)? How does the optimal batch size depend on the text size?
- The authors recommend a batch size of 8, because it captures most of the benefit. However, from their evaluation it seems that even higher batch sizes lead to even lower latency while quality stays the same for most models. Why is 16 not a better recommendation based on empirical evaluation?
-All tasks in the evaluation set are either binary classification tasks (sem_filter) or multi-class classification tasks with a fixed number of classes (up to 41 classes, sem_map). What about other or more complicated tasks, like extraction, ranking, summarization? Please add a discussion.

**Minor Comments:**

I really would like to see more tasks.

**Short Summary:**

In the paper, the authors investigate the optimal prompt-level batch size for Semantic Data Systems.
The prompt-level batch size is the number of data items (e.g. texts) that are processed per prompt
in the semantic data systems. Larger batch size usually results in faster processing, but previous
work avoided it due to potential loss in quality, citing the lost-in-the-middle effect.
The paper shows that in the cases they tested, there was almost no loss in quality when prompt-level
batch size increases. They recommend a batch size of 8.

**Strong Points:**

Strong:
- Low-latency semantic processing is a relevant problem. If increasing batch size does not lead to quality loss and only reduces latency, it's a low hanging fruit for improving existing systems.
- The model is justified both empirically and it is grounded in hardware.
- It is well written and easy to understand.

---

### Official Review · Reviewer_kzfg · 2026-07-10

**Confidence:** 4

**Improvement Opportunities:**

- The study focuses primarily on tuple-independent semantic filter and map operators, where cross-document interactions are minimal. It would strengthen the paper to evaluate whether batching benefits extends to other operators such as joins, aggregations, etc.
- The latency model and analysis rely on the assumption of the uniformity of API providers. A more explicit discussion of when these assumptions may not hold, such as under local inference, different serving engines, etc. would improve the robustness of the theoretical results.
- The recommendation of a default batch size of eight is appealing due to its simplicity, but it is derived from the evaluated workloads and models. Including additional experiments across more diverse datasets, longer-context applications, or open-weight models would also strengthen the claim that this recommendation generalizes beyond the current benchmark suite.

**Minor Comments:**

Some of the results presented in figure 2 should be included as a table. Other than that I think that this is a very legible paper.

**Short Summary:**

The paper investigates prompt level batching as a latency optimization for LLM-based semantic operators in database systems. The authors challenge the common assumption that batching multiple documents into a single prompt harms accuracy or increases latency due to the quadratic attention's nature. The authors performed experiments for five semantic processing tasks and five commercial LLMs, they demonstrate that prompt-level batching can reduce latency by up to 14× while maintaining virtually unchanged accuracy.They also propose a decomposed linear latency model that attributes the majority of execution time to fixed per-call overhead rather than token processing.

**Strong Points:**

- The paper evaluates the proposed batching strategy on five representative semantic processing tasks (including filtering and mapping) using five commercial LLMs from multiple providers. This experimental design strengthens the claim that the observed latency improvements generalize beyond a single model or benchmark.
- This work develops a decomposed latency model and performs a hardware-inspired analysis based on GPU memory and compute characteristics. This theoretical treatment provides intuition for when the proposed optimization should be effective and allows for an analytical recommendation for batch size.
- The proposed approach doesn't require changes to model architectures or serving infrastructure and can be easily adopted by semantic processing systems that rely on commercial LLM APIs
- The latency analysis as a linear model is also relevant to other applications beyond semantic operators

---

### Decision · Program_Chairs · 2026-07-16

**Decision:**

Accept

**Comment:**

This paper presents a timely and practical optimization for LLM-based semantic operators, showing that prompt-level batching can substantially reduce latency while preserving accuracy across several tasks and commercial models. The empirical evaluation and decomposed latency model provide clear guidance for system builders, including an actionable default batch size. We hope this work sparks useful discussion on efficient execution strategies for semantic data systems at the workshop.